# GEOMETRY-AWARE 6-DOF GRASP DETECTION IN COMPLEX SCENES

## ABSTRACT

Data-driven methods have made significant progress in 6-DoF grasp detection for robotic applications. However, reliably detecting grasps in cluttered scenes with transparent objects remains a challenge. To address this, we introduce TransCG-Grasp, an annotated extension of the TransCG dataset, to advance research in transparent object grasping. Additionally, we propose GA-Grasp, a novel geometry-aware 6-DoF grasp detection method designed to improve grasping for both transparent and general objects. GA-Grasp incorporates a modality-aware sparse tensor module and a geometry-aware sparse U-Net, leveraging RGB, depth, and surface normals to predict graspable points and generate final 6-DoF grasp poses. Extensive experiments on the TransCG-Grasp and GraspNet-1Billion datasets demonstrate that GA-Grasp outperforms existing methods. Notably, GA-Grasp surpasses the current state-of-the-art (SOTA) by an impressive margin of **10.06% AP** on the TransCG-Grasp dataset. In real-world experiments, our GA-Grasp achieves success rates of **82.0%** for transparent objects and **90.6%** for general objects, with a **100%** task completion rate, further validating its effectiveness for real-world robotic manipulation. The codes and trained models will be released upon acceptance.

## 1 INTRODUCTION

Intelligent robots play a pivotal role in various real-world applications, including picking Correll et al. (2016); Cao et al. (2023), assembling Sundermeyer et al. (2021), and cleaning Fu et al. (2024). Grasping is a fundamental task in robotic manipulation, encompassing perception, planning, and execution. During perception, grasp detection algorithms process input data to generate grasp poses, which are then fed into the robot control module to plan and execute trajectories for precise manipulation tasks, such as pick-and-place. Grasp detection methods are expected to handle objects with diverse shapes, sizes, appearances, materials, and poses. Traditional methods rely on manually designed policies and 3D object models Roa & Suárez (2009). However, their performance heavily depends on the accuracy of pose estimation and cannot generalize to unseen objects. With the advancement of deep learning, data-driven methods have achieved remarkable

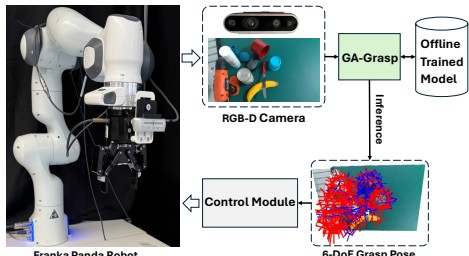

Figure 1: Overview of our system: An RGB-D camera mounted on the robot's wrist captures data of objects to be grasped. Our proposed GA-Grasp generates 6-DoF grasp poses during the inference process. These generated grasp poses are then utilized by the control module to plan and execute robot trajectories for pick-and-place tasks.

success in robotic grasp detection Lenz et al. (2015); Cao et al. (2021). Early approaches focused on predicting 4 degrees of freedom (DoF) planar grasps from single RGB-D observations Cao et al. (2021; 2023). However, these planar grasp methods constrain the manipulator to the workspace's normal direction, sacrificing DoF and limiting performance in complex scenarios.

Recently, 6-DoF grasp detection has gained significant attention because it enables robots to grasp objects from arbitrary orientations, making it ideal for cluttered and unstructured environments. Sampling-based methods Ten Pas et al. (2017); Liang et al. (2019) have been developed for 6-DoF

grasp detection, but their sampling-evaluation strategy is time-consuming. Due to limited data and corresponding grasp labels, researchers have proposed methods that learn under sparse supervision Qin et al. (2020); Liu et al. (2022); Wang et al. (2023); Breyer et al. (2021); Ni et al. (2020). However, the lack of diverse and high-quality labeled data restricts these models' ability to generalize effectively, resulting in poor grasp performance in real-world applications. To address this limitation, the GraspNet-1Billion dataset Fang et al. (2020) was introduced as a benchmark for training and evaluating 6-DoF grasp detection algorithms. Leveraging this dataset with dense labels, many 6-DoF grasp detection methods Fang et al. (2023; 2020); Gou et al. (2021); Ma & Huang (2023); Wang et al. (2021) have been proposed, achieving excellent performance through dense supervision. However, this approach introduces challenges, including significant resource costs and difficulties in learning and convergence during model training. To mitigate these resource costs, an economic framework for 6-DoF grasp detection was proposed in Wu et al. (2024).

Most existing methods are designed for general object grasping tasks with reliable depth observations, making them unsuitable for transparent object grasping. Commercial depth sensors often struggle to accurately capture and reconstruct depth maps for transparent and specular objects due to their inherent physical properties, such as reflection and refraction, which distort light paths and result in noisy, incomplete depth data. To address this challenge, some approaches leverage alternative cues. For instance, Ba et al. Ba et al. (2020) utilize polarization cues for shape estimation with a specialized polarization camera, while the authors of Li et al. (2020); Dai et al. (2023) employ multi-view images and material priors to reconstruct transparent object shapes. Other works Sajjan et al. (2020); Zhang & Funkhouser (2018); Wei et al. (2024); Zhu et al. (2021) adopt depth completion pipelines to estimate or restore depth information for transparent and specular objects. However, these methods rely on synthetic datasets, which fail to capture the real-world depth degradation caused by transparency. To bridge this synthetic-to-real gap, the TransCG dataset Fang et al. (2022) was introduced as a large-scale real-world dataset for transparent object depth completion, but it lacks grasp annotations. More recently, Shi et al. Shi et al. (2024) proposed leveraging raw IR observations from an active stereo camera to enhance depth estimation for transparent objects.

Despite these advancements, depth completion-based approaches for grasp detection remain two-stage processes: first, predicting accurate depth maps, and then backprojecting them into point clouds for SOTA point cloud-based grasp detection networks. This two-stage pipeline is suboptimal. An ideal framework for transparent object grasp detection should be optimized end-to-end for better performance and efficiency. To achieve this goal, a real-world transparent object dataset with grasp annotations is essential. However, collecting and labeling such a dataset is time-consuming and labor-intensive. In this work, we build upon the grasp annotation pipelines from GraspNet-1Billion Fang et al. (2020) and Contact GraspNet dataset Sundermeyer et al. (2021) to annotate the TransCG dataset Fang et al. (2022), resulting in TransCG-Grasp. Furthermore, we propose GA-Grasp, a novel geometry-aware 6-DoF grasp detection framework designed to enhance transparent and general object grasping. GA-Grasp consists of a modality-aware sparse tensor module, a geometry-aware sparse U-Net, and a grasp head. To leverage multi-modality cues, RGB color and geometric information (depth and surface normals) are fused into sparse tensors. The geometry-aware sparse U-Net is used to directly predict graspable points. This approach differs from current methods like GSNet Wang et al. (2021) and EconomicGrasp Wu et al. (2024), which rely on graspness to identify graspable points from input point clouds. These methods require setting predefined thresholds, introducing complexity, and making it difficult to determine an optimal threshold. In contrast, GA-Grasp eliminates the need for graspness scores, allowing it to directly learn graspable points without threshold dependency. Once graspable points are identified, the grasp head predicts the final 6-DoF grasp poses. Experimental results on the TransCG-Grasp and GraspNet-1Billion datasets demonstrate that GA-Grasp outperforms current grasp detection methods. Notably, GA-Grasp achieves a **10.06%** improvement in overall AP on the TransCG-Grasp dataset, significantly surpassing the second-best method, EconomicGrasp Wu et al. (2024). As shown in Fig. 1, we integrate GA-Grasp into a real robotic system for pick-and-place tasks. This system comprises a Franka Panda robot, a wrist-mounted RGB-D camera, the GA-Grasp detection model, and a control module for execution. In real-world grasping experiments, GA-Grasp achieves high grasp success rates (**82.0%** for transparent objects and **90.6%** for general objects) and completion rates of **100%** for both tasks, further validating its practical effectiveness for robotic manipulation.

The main contributions of this work can be summarized as follows:

- To advance research in transparent object grasping, we annotate the TransCG dataset, creating the TransCG-Grasp dataset.

- We propose GA-Grasp, a novel geometry-aware 6-DoF grasp detection method composed of a modality-aware sparse tensor module, a geometry-aware sparse U-Net, and a grasp head.

- Extensive experiments demonstrate that our GA-Grasp outperforms existing approaches on both TransCG-Grasp and GraspNet-1Billion datasets. When deployed in a real robotic system for pick-and-place tasks, GA-Grasp achieves high grasp success rates (**82.0%** for transparent objects and **90.6%** for general objects) and **100%** completion rates for both tasks, further validating its effectiveness for robotic manipulation.

## 2 RELATED WORK

**Transparent object dataset.** For tasks such as transparent object classification, segmentation, and pose estimation, depth information is not strictly necessary. Large-scale datasets such as Trans10K-V2 Xie et al. (2021), Stanford2D-3D Armeni et al. (2017), and StereOBJ-1M Fu et al. (2020) have been collected for these tasks. However, for tasks where depth information is crucial, researchers often rely on synthetic datasets as a solution. Examples include the ClearGrasp synthetic dataset Sajjan et al. (2020) and the Omniverse object dataset Zhu et al. (2021), both generated using SuperCaustics Mousavi & Estrada (2021). Despite their usefulness, these synthetic datasets exhibit a domain gap between their depth data and the real-world depth captured by commercial depth sensors. To mitigate this gap, real-world datasets such as TODD Xu et al. (2021) for depth completion and a keypoint estimation dataset Liu et al. (2020) for transparent objects have been introduced. Additionally, the TransCG dataset Fang et al. (2022) was developed as a large-scale real-world dataset for transparent object depth completion. However, TransCG lacks grasp annotations, limiting its application in transparent object grasping research. In this work, we introduce TransCG-Grasp, an annotated extension of the TransCG dataset, designed to advance research in transparent object grasping.

**6-DoF grasp detection methods.** 6-DoF grasp detection is the task of predicting the position and orientation of the gripper in 3D space. By enabling robots to grasp objects from various angles, it serves as the foundation for robotic manipulation. In Ten Pas et al. (2017), the authors propose GPD, a two-stage 6-DoF grasping method that estimates grasp candidates sampled under empirical constraints. PointNetGPD Liang et al. (2019) improves upon GPD by leveraging PointNet Qi et al. (2017) for grasp evaluation. Furthermore, grasp poses are directly regressed from partial viewpoint clouds in S4G Qin et al. (2020) and through PointNet++-extracted features in Ni et al. (2020). Researchers have also proposed methods that learn under sparse supervision Liu et al. (2022); Wang et al. (2023); Breyer et al. (2021). To address the lack of diverse and high-quality labeled data, the GraspNet-1Billion dataset for general object grasping and a grasp pose prediction network were introduced in Fang et al. (2020). By incorporating RGB and depth information as input, the authors of Gou et al. (2021) enhance the performance of 6-DoF grasping. A geometrically based graspness is introduced in Wang et al. (2021) to predict graspable points in cluttered scenes. HGGD Chen et al. (2023) employs 2D CNNs to generate heatmaps for grasp locations, integrating 2D semantics with 3D geometric features to predict 6-DoF grasp poses. More recently, EconomicGrasp Wu et al. (2024) proposes an efficient framework to reduce resource costs in model training. Unlike the methods mentioned above, we propose a novel 6-DoF grasp detection method designed to improve grasp detection performance for both transparent and general objects.

## 3 METHOD

### 3.1 OVERVIEW

As illustrated in Fig. 2, our GA-Grasp framework comprises three key components: a modality-aware sparse tensor module, a geometry-aware sparse U-Net, and a grasp head. First, we leverage the sparse tensor mechanism to efficiently integrate RGB color information with geometric features, including depth and surface normals. The generated sparse tensors are then processed by the geometry-aware sparse U-Net to extract meaningful features. Notably, the pruning layer in the decoder removes background points while preserving target graspable points from the input sparse

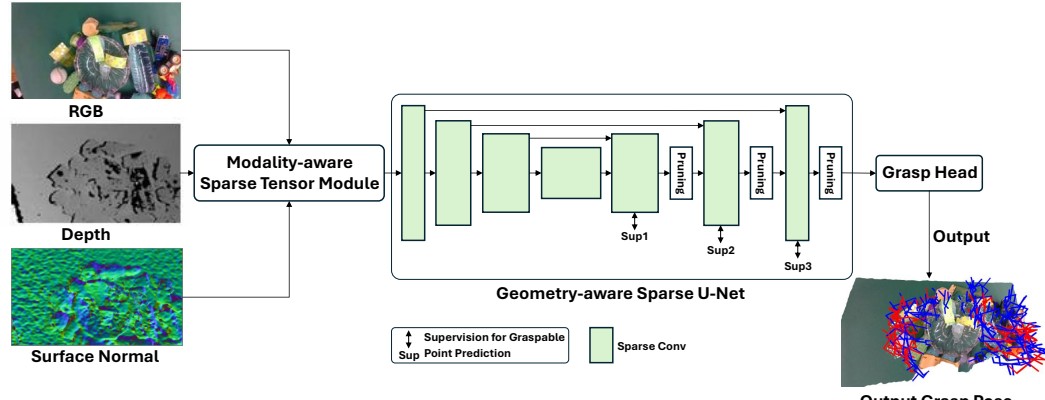

Figure 2: Overall pipeline of GA-Grasp: our proposed method consists of a modality-aware sparse tensor module, a geometry-aware sparse U-Net, and a grasp head. First, the model converts multimodal input data into sparse tensors. These tensors are then passed through the geometry-aware sparse U-Net, which extracts features for two tasks: graspable point prediction and grasp pose prediction tasks.

tensors. GA-Grasp is designed as an end-to-end jointly learned framework, where the extracted features are utilized for both graspable points prediction and grasp pose prediction tasks. In the following parts, we provide a detailed explanation of our proposed method.

## 3.2 MODALITY-AWARE SPARSE TENSOR MODULE

RGB images are commonly used as input data for grasp detection tasks due to their rich color features, such as hue, saturation, and brightness, which help identify objects. However, challenges such as lighting variations and occlusions can lead to inconsistent color representations, potentially degrading algorithm performance. Depth data, on the other hand, provides crucial spatial information about the distances between objects, enabling systems to distinguish between objects that may appear similar in RGB images but are located at different positions. Despite its advantages, depth data is susceptible to environmental factors, such as reflective or transparent surfaces, which can compromise accuracy. To address these limitations, we introduce surface normals into the grasp detection framework. Surface normals encode the local geometry of objects, offering valuable insights into surface orientation. While RGB data captures color and texture, and depth data provides spatial position, surface normals complement these modalities by describing surface geometry. By integrating these three modalities—RGB, depth, and surface normals—our approach achieves a more comprehensive understanding of the scene, enhancing grasp detection accuracy. Based on the refined depth data and camera parameters, we generate point clouds and randomly sample a fixed set of 20,000 points. This sampling strategy ensures computational efficiency while preserving essential data features. Next, the 3D space is partitioned into discrete cubic cells with a voxel size of 0.005 meters during the voxelization process. This voxel size, chosen based on prior work Wu et al. (2024), strikes a balance between detail and computational efficiency. To efficiently integrate RGB color and geometric information (depth and surface normals), we avoid complex fusion methods and instead employ the sparse tensor mechanism Choy et al. (2019). This approach transforms the inputs into unique coordinates and associated features, which are then processed by the sparse convolutional neural network to extract valuable features for subsequent tasks. This design enables GA-Grasp to achieve superior performance in complex scenes.

## 3.3 GEOMETRY-AWARE SPARSE U-NET

**Backbone.** We build our geometry-aware sparse U-Net using the sparse convolutional neural network developed in MinkowskiEngine Choy et al. (2019). As shown in Fig. 2, sparse U-Net follows an encoder-decoder architecture, where the encoder downsamples input sparse tensors and maps them to a latent feature space, while the decoder upsamples these latent features into high-resolution

representations with deep semantic information. To mitigate information loss from downsampling and upsampling operations, skip connections are introduced between corresponding encoder and decoder layers. Specifically, the encoder consists of layers with channels of 32, 64, 128, and 256, while the decoder has layers with channels of 256, 192, 192, and 256. To ensure consistent channel dimensions between the encoder and decoder, we incorporate $1\times1$ convolutions in the skip connections.

**Graspable point prediction.** Previous works Wang et al. (2021); Wu et al. (2024) rely on graspness to identify graspable points from the input data. However, these approaches require predicting graspness scores and selecting points that exceed a predefined threshold, introducing complexity and making it difficult to determine an optimal threshold. To address these limitations, we eliminate graspness from our grasp detection framework and instead employ supervision learning to directly predict graspable points. As presented in Fig. 2, we use three-level supervision, where pruning layers in the decoder remove background points while preserving target graspable points. By bypassing the need for graspness scores, our method learns graspable points in an end-to-end manner. Once graspable points are predicted, we apply furthest point sampling (FPS) to maximize the distances among sampled points. This process selects $M$ seed points, each represented by $(3 + C)$-dim features, where 3 corresponds to point coordinates and $C$ denotes the feature embeddings output by the sparse U-Net.

### 3.4 GRASP HEAD

Following previous works Wang et al. (2021); Wu et al. (2024), we apply the multi-layer perceptron (MLP) to the sampled seed points to generate two outputs: $M \times C$ residual features for grasp generation and $M \times V$ vectors for view-wise graspable landscapes. The locations of the cylinder spaces are determined by the coordinates of the seed points, while their directions are defined by the view vectors. We utilize the focal representation module from Wu et al. (2024) to produce the final outputs. The output size is $M \times (A \times D \times 2)$, where $A$ represents the number of in-plane rotation angles, $D$ represents the number of gripper depths, and 2 represents the score and width.

### 3.5 LOSS FUNCTION

The graspable point prediction and grasp pose prediction tasks are trained simultaneously. Specifically, we define the multi-task loss as follows:

$$L = \alpha L_{gp} + \beta L_v + \gamma(L_a + L_d + L_s) + \lambda L_w. \tag{1}$$

where $L_{gp}$, $L_v$, $L_a$, $L_d$, $L_s$, and $L_w$ represent graspable point prediction loss, view-wise graspable landscape loss, in-plane rotation angle loss, gripper depth loss, grasp scores loss, and grasp width loss, respectively. The terms $\alpha, \beta, \gamma, \lambda$ are their corresponding weighting coefficients. Specifically, graspable point prediction loss is calculated by using BCEWithLogitsLoss. CrossEntropyLoss is used for in-plane rotation angle loss, gripper depth loss, and grasp scores loss. And we use Smooth-$L_1$ loss for regression tasks, including view-wise graspable landscape loss and grasp width loss.

## 4 EXPERIMENTS

### 4.1 DATASETS

**TransCG-Grasp dataset.** The TransCG dataset Fang et al. (2022) is a real-world dataset that provides ground-truth depth, surface normals, and transparent object masks in diverse and cluttered scenes. However, since it does not include grasp annotations, we introduce TransCG-Grasp, a labeled dataset tailored for transparent object grasping. The original TransCG dataset contains 130 scenes: 65 scenes with only transparent objects and 65 scenes with transparent objects mixed with normal diffusive objects. However, the 3D mesh models for the normal diffusive objects are unavailable, making grasp annotation infeasible. As a result, we exclude these 65 mixed scenes. Additionally, the objects in the TransCG dataset are attached with optical markers, which interfere with grasp annotations. To address this, we remove the markers from the object models. For annotation, we follow the grasp annotation pipeline from GraspNet-1Billion Fang et al. (2020) and

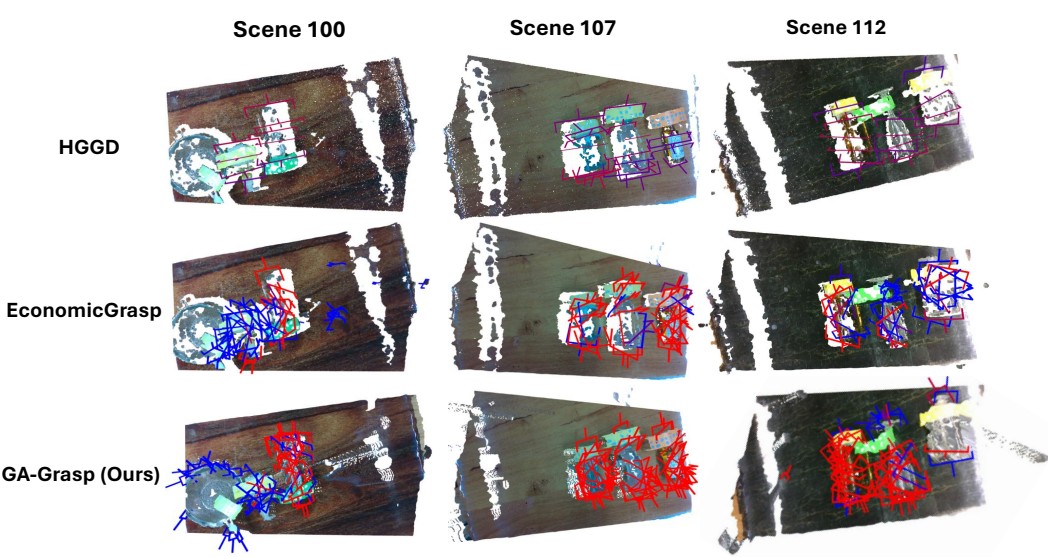

Figure 3: Qualitative results on the TransCG-Grasp dataset. The prediction outputs from HGGD, EconomicGrasp, and our GA-Grasp are visualized. Specifically, the top 50 grasp candidates after applying grasp-NMS Fang et al. (2020) are visualized. The color gradient represents the predicted grasp confidence, with red indicating high confidence and blue indicating low confidence.

Contact GraspNet dataset Sundermeyer et al. (2021), which consists of three key steps: data acquisition (TransCG data), object-level grasp generation, and scene-level grasp generation. This pipeline is designed to annotate cluttered grasp scenes efficiently. To ensure compatibility and uniformity, we reformat all data to align with the GraspNet-1Billion Fang et al. (2020) dataset structure. For the labeled data, we split the dataset into 35 scenes for training and 30 scenes for testing. More details can be found in the **Appendix**.

**GraspNet-1Billion dataset.** GraspNet-1Billion Fang et al. (2020) is a large-scale dataset that provides densely annotated cluttered scenes featuring seen, similar, and novel objects. The dataset includes 88 objects with high-quality 3D mesh models, ensuring geometric diversity. Specifically, 13 adversarial objects are selected from DexNet 2.0 Mahler et al. (2017), 43 objects are unique to GraspNet-1Billion, and 32 objects are chosen from the YCB dataset Calli et al. (2015). To capture the scenes, two widely used RGB-D cameras—Intel RealSense 435 and

Table 1: Performance comparison on the TransCG-Grasp dataset. The best results are highlighted in bold.

| Methods | AP % | $AP_{0.8}$ % | $AP_{0.4}$ % |
|---|---|---|---|
| GraspNet Fang et al. (2020) | 7.85 | 9.64 | 4.04 |
| GSNet Wang et al. (2021) | 12.08 | 15.18 | 6.45 |
| TSB Ma & Huang (2023) | 13.23 | 16.65 | 7.35 |
| TransCG Fang et al. (2022) | 19.51 | 24.58 | 11.15 |
| HGGD Chen et al. (2023) | 19.25 | 24.42 | 12.31 |
| EconomicGrasp Wu et al. (2024) | 19.67 | 24.81 | 11.35 |
| GA-Grasp (Ours) | **29.73** | **37.31** | **21.08** |

Azure Kinect—were employed to record data simultaneously. For each scene, 8 to 12 objects were randomly selected and arranged in a cluttered manner. Each viewpoint includes a synchronized image pair from both cameras, along with corresponding camera poses. GraspNet-1Billion contains 48,640 images per camera type, covering a total of 190 scenes, with 100 scenes for training and 90 for evaluation. The evaluation scenes are divided into three categories: seen, similar, and novel.

## 4.2 IMPLEMENTATION DETAILS

Our models are implemented using PyTorch Paszke et al. (2019) and the Minkowski Engine Choy et al. (2019). For optimization, we employ the Adam optimizer Kingma & Ba (2014) with an initial learning rate of $1e^{-3}$ and a cosine learning rate decay schedule. The training is conducted with a batch size of 4 for 10 epochs, and all models are trained on an NVIDIA RTX 4090 GPU. For evaluation, we adopt the Average Precision (AP) metric.

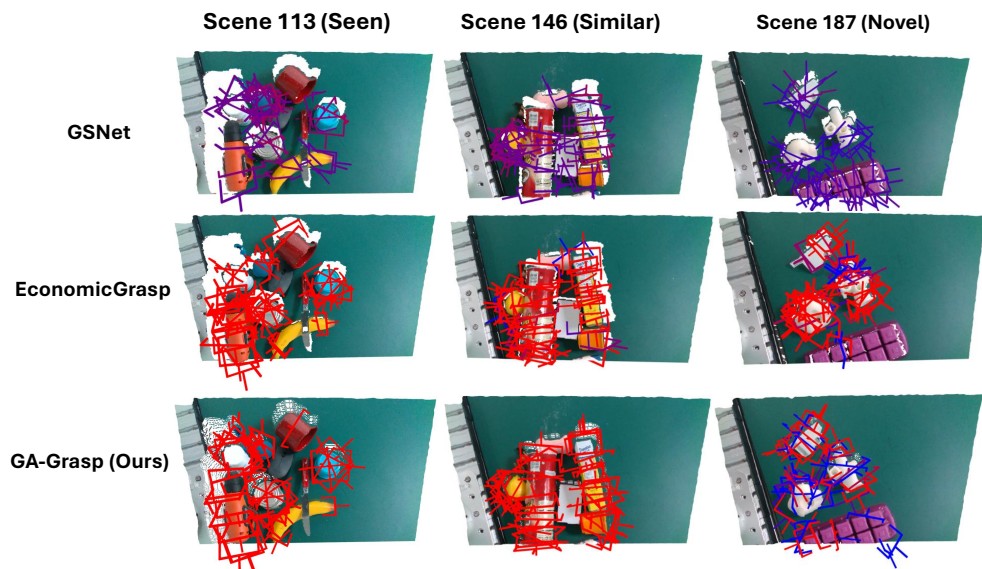

Figure 4: Qualitative results on the GraspNet-1Billion dataset. The prediction outputs from GSNet, EconomicGrasp, and our GA-Grasp are visualized across three test scenes: seen, similar, and novel. Specifically, the top 50 grasp candidates after applying grasp-NMS Fang et al. (2020) are visualized. The color gradient indicates the predicted grasp confidence, with red representing high confidence and blue representing low confidence.

### 4.3 RESULTS ON THE TRANSCG-GRASP DATASET

To further validate the effectiveness of our proposed method, we conduct experiments for transparent object grasping on our labeled TransCG-Grasp dataset. For a comprehensive comparison, we evaluate our method against several open-sourced alternatives, including GraspNet Baseline Fang et al. (2020), TSB Ma & Huang (2023), GSNet Wang et al. (2021), HGGD Chen et al. (2023), and EconomicGrasp Wu et al. (2024). To ensure a fair assessment, all models are trained on the same hardware device and under identical experimental settings. The results are presented in Tab. 1. Our GA-Grasp outperforms the second-best approach, EconomicGrasp Wu et al. (2024), with grasp detection accuracy of **29.73% AP** on the TransCG-Grasp dataset. Notably, our GA-Grasp achieves a **10.06% AP** improvement over EconomicGrasp in terms of AP, demonstrating its superior ability to understand transparent object grasping scenarios. As shown in Fig. 3, we present the grasp detection results of HGGD Chen et al. (2023), EconomicGrasp Wu et al. (2024), and our GA-Grasp on the TransCG-Grasp dataset. The top 50 grasp candidates after applying grasp non-maximum suppression (grasp-NMS) Fang et al. (2020) are visualized. Grasp-NMS is employed to filter out low-quality grasp poses, ensuring that only high-quality grasps are selected for execution. After Grasp-NMS, HGGD generates limited grasp candidates caused by transparent objects. In contrast, GA-Grasp predicts a rich set of grasp candidates, even in challenging transparent object scenes. By leveraging multi-modality data (RGB, depth, and surface normal) and the proposed grasp point prediction method, GA-Grasp achieves excellent performance in scenarios with transparent objects. This capability is crucial for real-world robotic manipulation, where reliable grasping of diverse objects is essential. Overall, our qualitative evaluation highlights the effectiveness of GA-Grasp in addressing transparent object challenges and reinforces its potential to enhance grasping techniques for more advanced robotic applications.

### 4.4 RESULTS ON THE GRASPNET-1BILLION DATASET

We further conduct a comprehensive comparative analysis on the GraspNet-1Billion dataset, evaluating our method against several SOTA 6-DoF grasping approaches, including GPD Ten Pas et al. (2017), PointNetGPD Liang et al. (2019), S4G Qin et al. (2020), TransGrasp Liu et al. (2022), GraNet Wang et al. (2023), GraspNet Baseline Fang et al. (2020), TSB Ma & Huang (2023),

Table 2: Performance comparison on the GraspNet-1Billion dataset Fang et al. (2020) for both RealSense and Kinect cameras. The best results are highlighted in bold.

| Supervision | Methods | Average / AP % | Seen / AP % | Similar / AP % | Novel / AP % |
|---|---|---|---|---|---|
| Sample | GPD Ten Pas et al. (2017) | 17.48/19.05 | 22.87/24.38 | 21.33/23.18 | 8.24/9.58 |
| | PointNetGPD Liang et al. (2019) | 19.29/20.88 | 25.96/27.59 | 22.68/24.38 | 9.23/10.66 |
| Sparse | S4G Qin et al. (2020) | 17.73/11.97 | 25.71/18.72 | 18.45/11.82 | 9.04/5.38 |
| | TransGrasp Liu et al. (2022) | 27.65/25.70 | 39.81/35.97 | 29.32/29.71 | 13.83/11.41 |
| | GraNet Wang et al. (2023) | 32.74/29.41 | 43.33/41.38 | 39.98/35.29 | 14.90/11.57 |
| Dense | GraspNet Baseline Fang et al. (2020) | 21.41/23.08 | 27.56/29.88 | 26.11/27.84 | 10.55/11.51 |
| | TSB Ma & Huang (2023) | 44.85/35.42 | 58.95/49.42 | 52.97/41.49 | 22.63/15.35 |
| | GSNet Wang et al. (2021) | 47.81/42.53 | 65.70/61.19 | 53.75/47.39 | 23.98/19.01 |
| Economic | HGGD Chen et al. (2023) | 42.79/39.79 | 58.35/56.85 | 47.93/43.93 | 22.10/18.59 |
| | EconomicGrasp Wu et al. (2024) | 51.63/44.62 | 68.21/62.59 | 61.19/**51.73** | 25.48/19.54 |
| | GA-Grasp (Ours) | **53.62/47.07** | **71.46/68.67** | **62.04**/51.48 | **27.35/21.06** |

Table 3: Ablation studies of different input modalities on the TransCG-Grasp and GraspNet-1Billion datasets Fang et al. (2020) (tested on the Realsense Split).

| RGB | Depth | Surface Normal | TransCG-Grasp | | | GraspNet-1Billion | | | |
|---|---|---|---|---|---|---|---|---|---|
| | | | AP % | AP$_{0.8}$ % | AP$_{0.4}$ % | Average / % | Seen / AP % | Similar / AP % | Novel / AP % |
| ✓ | ✓ | | 29.24 | 36.80 | 20.62 | 52.31 | 69.99 | 60.46 | 26.47 |
| | ✓ | ✓ | 28.77 | 36.08 | 20.68 | 52.07 | 69.30 | 61.91 | 25.01 |
| ✓ | ✓ | ✓ | **29.73** | **37.31** | **21.08** | **53.62** | **71.46** | **62.04** | **27.35** |

GSNet Wang et al. (2021), HGGD Chen et al. (2023), and EconomicGrasp Wu et al. (2024). The experimental results for both RealSense and Kinect cameras are summarized in Tab. 2. Our GA-Grasp achieves an overall average performance of **53.62%/47.07% AP** with RealSense/Kinect cameras, surpassing all other methods. Moreover, our GA-Grasp consistently achieves performance improvement across both the TransCG-Grasp and GraspNet-1Billion datasets, showcasing its effectiveness and robust performance in complex, real-world scenes. The smaller improvement on the GraspNet-1Billion dataset compared to TransCG is due to the fact that, for general objects, depth sensors can provide sufficient depth information to construct accurate point clouds. Therefore, our method does not exhibit a significant advantage in such cases. Additionally, we provide qualitative results by visualizing grasp predictions across three test scenarios: seen, similar, and novel. As shown in Fig. 4, GA-Grasp generates high-quality grasp detections with higher confidence compared to previous methods like GSNet Wang et al. (2021) and EconomicGrasp Wu et al. (2024). These qualitative results further demonstrate the effectiveness of our approach in cluttered environments with occlusion and incomplete observations.

## 4.5 ABLATION STUDY

We conduct ablation studies on the TransCG-Grasp and GraspNet-1Billion datasets. Specifically, we analyze the effects of different input modalities. The experimental results are summarized in Tab. 3. We train our model with different input modalities under the same experimental settings. The results show that combining all modalities, RGB, depth, and surface normal, achieves the best grasp detection performance, **29.73% AP** on the TransCG-Grasp dataset and **53.62% average AP** on the GraspNet-1Billion dataset, respectively. Without each modality RGB or surface normal, the grasp detection performance decreases. These results demonstrate the effectiveness of leveraging both RGB-based and geometric-based features, such as depth and surface normals, in 6-DoF grasp detection tasks.

## 4.6 GRASPING EXPERIMENTS IN THE REAL WORLD

To further validate the effectiveness of our proposed GA-Grasp, we conduct real-world grasping experiments for both transparent and general object grasping tasks. As illustrated in Fig. 5, our hardware setup consists of a Franka Panda robotic arm, a Robotiq Adaptive 2F-85 two-finger gripper, and an Intel RealSense D435i RGB-D camera. We evaluate our method using a diverse set of real-world objects, including both transparent and opaque items, to assess its performance across

Table 4: Real-world experiments for both transparent object and general object grasping tasks.

| Scene | Transparent object grasping | | | | General object grasping | | | |
|---|---|---|---|---|---|---|---|---|
| | Objects | Attempts | Success Rate | Completion Rate | Objects | Attempts | Success Rate | Completion Rate |
| 1 | 7 | 9 | 77.8% | 100% | 7 | 7 | 100% | 100% |
| 2 | 7 | 8 | 87.5% | 100% | 7 | 8 | 87.5% | 100% |
| 3 | 7 | 9 | 77.8% | 100% | 7 | 7 | 100% | 100% |
| 4 | 7 | 10 | 70.0% | 100% | 6 | 8 | 75% | 100% |
| 5 | 6 | 6 | 100% | 100% | 6 | 7 | 85.7% | 100% |
| 6 | 8 | 11 | 72.7% | 100% | 8 | 8 | 100% | 100% |
| 7 | 8 | 8 | 100% | 100% | 7 | 8 | 87.5% | 100% |
| Total | **50** | **61** | **82.0%** | **100%** | **48** | **53** | **90.6%** | **100%** |

different objects. By deploying the GA-Grasp model on the robotic system, the robot executes grasp attempts based on the predicted 6-DoF grasp poses for pick-and-place tasks. We measure success rates (objects / attempts) and completion rates (successfully cleared scene number / scene number) under various conditions. We also include video demonstrations and additional details in the **Appendix**.

**Transparent object grasping experiments.** Transparent objects present significant challenges due to the lack of reliable depth information. To evaluate the proposed GA-Grasp, we test it on a diverse set of transparent objects in cluttered scenes. Specifically, we arrange 7 test scenes containing 7, 7, 7, 7, 6, 8, and 8 objects, respectively. The experimental results, shown in Tab. 4, demonstrate that GA-Grasp achieves an excellent grasp success rate of **82.0%** and a completion rate of **100%**. These results highlight the effectiveness of our GA-Grasp in handling transparent object grasping.

**General object grasping experiments.** Similarly, we arrange 7 scenes with 7, 7, 7, 6, 6, 8, and 7 objects for general object grasping experiments. As shown in Tab. 4, our GA-Grasp achieves a high grasp success rate of **90.6%** and a completion rate of **100%**, demonstrating its effectiveness in general object grasping tasks. These results further validate the versatility of our method across diverse object types and scenarios.

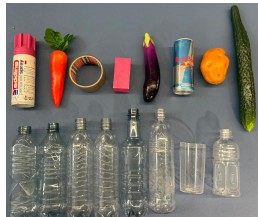 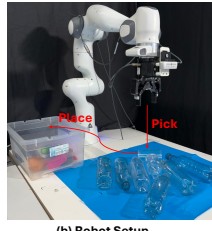

(a) Object Samples      (b) Robot Setup

Figure 5: Real-world object samples and robot setup.

The real-world grasping experiments confirm that the proposed GA-Grasp effectively handles both transparent and general object grasping tasks, making it a promising solution for real-world robotic manipulation. However, while the success rate for general object grasping is high, the performance for transparent object grasping is comparatively lower. This highlights the need for further development of novel grasp detection methods to address challenges posed by transparent objects, particularly due to the lack of depth information.

## 5 CONCLUSION

To address the research gap in transparent object grasping, we introduce the TransCG-Grasp dataset, a labeled version of the TransCG dataset Fang et al. (2022). Additionally, we propose a novel geometry-aware 6-DoF grasp detection framework, GA-Grasp. By leveraging multi-modality data (RGB, depth, and surface normal) and a novel grasp point prediction mechanism, GA-Grasp achieves excellent performance in both transparent object and general object grasping tasks. Extensive experiments on the TransCG-Grasp and GraspNet-1Billion datasets demonstrate the effectiveness of the proposed GA-Grasp. Our method outperforms current SOTA approaches both qualitatively and quantitatively. Notably, GA-Grasp achieves an overall performance of **29.73% AP** on the TransCG-Grasp dataset, significantly surpassing the second-best method, EconomicGrasp, which achieves **19.67% AP**. In real-world grasping experiments, GA-Grasp achieves high grasp success rates (**82.0%** for transparent objects and **90.6%** for general objects) and completion rates (**100%** for both tasks), further validating its practical viability for robotic manipulation.

ETHICS STATEMENT

This research adheres to the ethical standards of the ICLR community. All datasets used in our experiments are publicly available and contain no personally identifiable or sensitive information. Our models are developed solely for academic research purposes. We recognize that grasp detection techniques could potentially be applied in sensitive domains (e.g., autonomous robotics, military), and we encourage their responsible use. We explicitly oppose harmful exploitation and strongly advocate for strict governance frameworks to ensure responsible development and deployment, minimizing potential societal risks.

REPRODUCIBILITY STATEMENT

We will release the full code, configurations, preprocessing and evaluation scripts, and our trained weights upon acceptance.

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

# A  APPENDIX

## A.1  USE OF LLMS

We used LLMs only for grammar, wording, and formatting edits. All technical content, analyses, and reported results were authored and verified by the authors. There is no scientific claims or data that were generated by the LLMs.

## A.2  TRANSPARENT OBJECT GRASP ANNOTATION

Previous datasets for transparent objects primarily focus on depth completion or pose estimation, lacking 6-DoF grasp annotations. To enable end-to-end training of a 6-DoF grasp detection network for transparent objects, we introduce the TransCG-Grasp dataset, a labeled extension of the TransCG dataset Fang et al. (2022). The grasp annotation process is divided into three steps: data acquisition, object-level grasp generation, and scene-level grasp generation. In this section, we provide further details on each step.

### A.2.1  DATA ACQUISITION

We utilize data from the TransCG dataset Fang et al. (2022). The objects in the TransCG dataset vary in shape, texture, size, material, and other attributes. However, these objects are equipped with markers, which interfere with grasp annotations. To eliminate the influence of optical markers, we remove them from the object models.

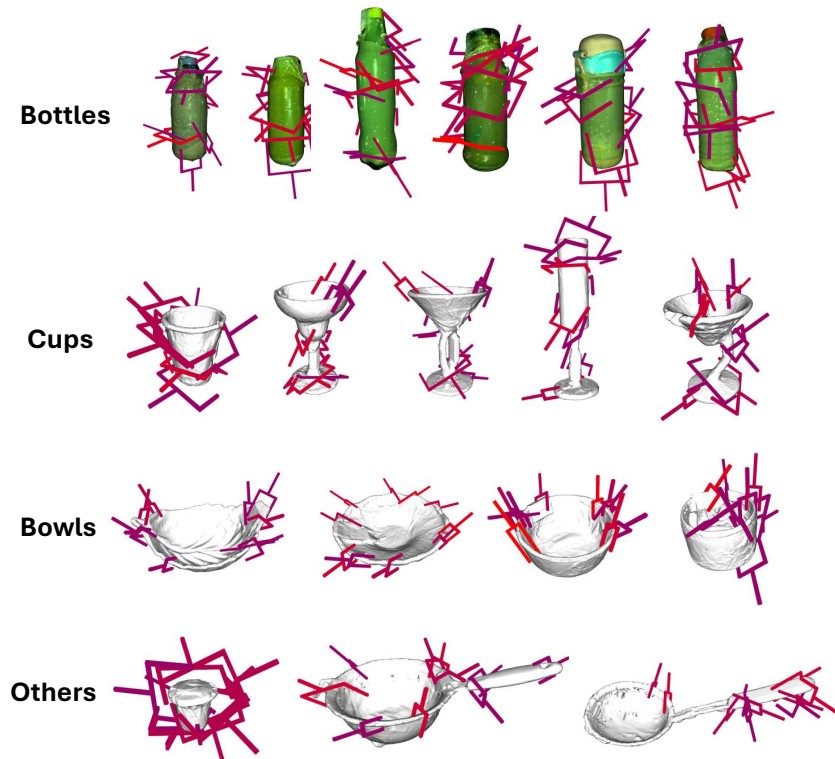

Figure 6: Annotation samples selected from the TransCG-Grasp dataset. For clear visualization of objects, only 10 grasp labels are plotted.

### A.2.2 OBJECT-LEVEL GRASP GENERATION

Unlike labels in common vision tasks, 6-DoF grasp poses span a broad and continuous search space, resulting in an infinite number of potential annotations. Manually annotating every scene would require an impractical amount of effort. Given that all objects are known, we follow the automated workflow from GraspNet-1Billion Fang et al. (2020) and Contact GraspNet dataset Sundermeyer et al. (2021) for grasp position annotation. Initially, grasp positions are generated and labeled for each individual object. This is achieved by downsampling high-quality mesh models so that the sampled points, referred to as grip points, are evenly distributed in voxel space. For each grasp point, we sample $V$ views uniformly distributed across a spherical space. A two-dimensional grid, $D \times A$, is used to search for grasp candidates, where $D$ represents the set of gripper depths and $A$ represents the set of in-plane rotation angles. The gripper width is selected to avoid empty grasps and collisions. Based on the mesh model, each grasp candidate is assigned a confidence score. To evaluate each grasp, we use an analytic computation approach. The force-closure metric Nguyen (1988); Ten Pas et al. (2017), effective for grasp evaluation, produces a binary label indicating whether the grasp is antipodal under a given friction coefficient $\mu$. Given a grasp pose, the associated object, and the metric, the result is robustly computed based on physical principles. In this work, we use an enhanced metric as described in Liang et al. (2019). We incrementally increase $\mu$ from 0.1 to 1 in steps of $\Delta\mu = 0.1$ until the grasp becomes antipodal. A lower friction coefficient $\mu$ indicates a higher likelihood of grasp success. Consequently, the score $s$ is defined as:

$$s = 1.1 - \mu. \tag{2}$$

where $s$ lies in the range $(0, 1)$. As shown in Fig. 6, we visualize 10 grasp labels for selected object samples from the TransCG-Grasp dataset. These include transparent and shiny objects such as bottles, cups, and bowls made of glass and plastic, as well as other items like pans.

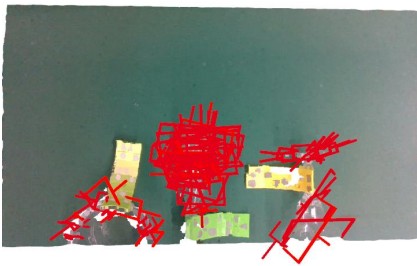 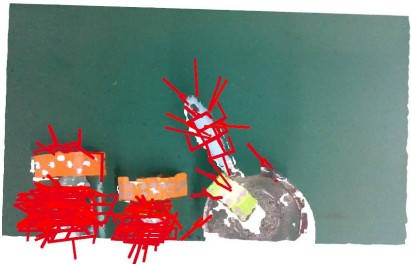

(a) Scene containing transparent objects.   (b) Scene containing transparent and reflective objects.

Figure 7: Our annotated scenes with grasp labels.

| Sparse U-Net Layers | AP | $AP_{0.8}$ | $AP_{0.4}$ |
|---|---|---|---|
| 3 | 68.94 | 81.12 | 63.57 |
| 4 | **71.46** | **83.17** | **66.77** |

Table 5: Ablation studies of different sparse U-Net layers on the GraspNet-1Billion dataset Fang et al. (2020). All experiments are tested on Realsense Split (Seen).

### A.2.3    SCENE-LEVEL GRASP GENERATION

**Operation plane fitting by 3D points.**    In the GraspNet-1Billion Fang et al. (2020) dataset, a 3D table model is used to check for object grasp collisions with the table surface. To simplify this process, we approximate the table as an infinite plane. For scene-level grasp annotation, we first determine the operation plane, where objects are placed, to filter out grasps that collide with the plane. A plane is mathematically defined by a normal vector $n = [a, b, c]^T$ and a distance $d$, such that any point $p = [x, y, z]^T$ on the plane satisfies the equation:

$$np + d = 0. \tag{3}$$

Expanding this, we obtain:

$$ax + by + cz + d = 0. \tag{4}$$

The equation is overdetermined because, while the solution space (a plane) is three-dimensional, the explanation given above involves four parameters. To reduce the degrees of freedom (DoF), we impose a constraint by setting $c = 1$, meaning the z-component of the plane normal is always fixed to one (note that the normal does not need to be unit length). By applying this constraint, the plane equation simplifies to:

$$ax + by + z + d = 0. \tag{5}$$

**Scene grasp annotation.**    The pipeline for scene-level grasp annotation involves several steps. First, we calculate the object poses relative to the camera coordinate system. Next, we transform the object grasps into the scene using these object poses. We then remove grasps that collide with the plane or other objects, as well as those that fall outside the image plane. Finally, we obtain the scene-level grasps. Fig. 7 presents grasp samples from our annotated transparent grasp dataset, demonstrating the effectiveness of our annotation pipeline.

### A.3    MORE EXPERIMENTAL ANALYSIS

### A.3.1    IMPACT OF DIFFERENT SPARSE U-NET LAYERS

Using RGB, depth, and surface normal as inputs, we conduct experiments on the GraspNet-1Billion dataset to explore the impact of the number of sparse U-Net layers. As shown in Tab. 5, a sparse U-Net with 4 layers outperforms one with 3 layers. Increasing the number of layers in the U-Net architecture allows the model to extract deeper features from the input data, enabling it to learn richer semantic representations and leading to improved grasp detection performance. These results highlight the importance of architectural depth in enhancing feature extraction capabilities.

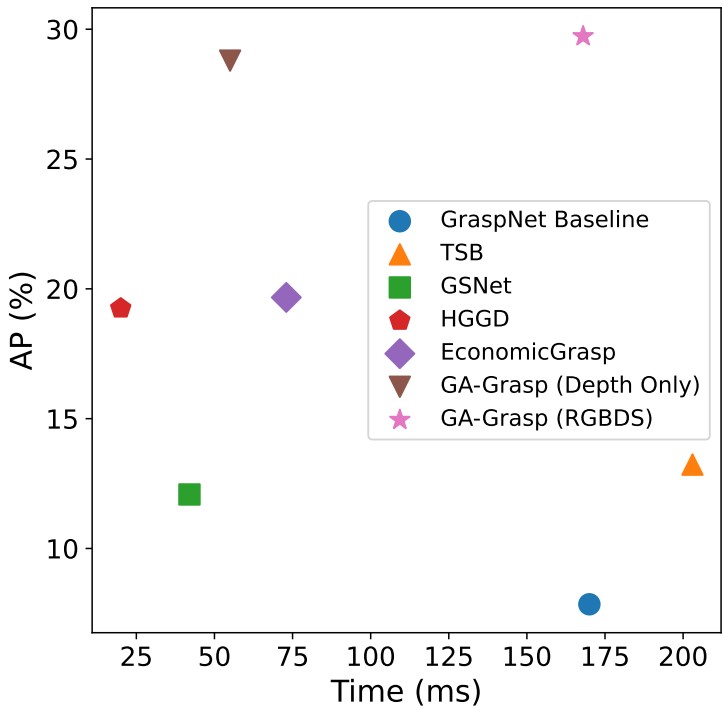

Figure 8: AP versus time curves for different methods on the TransCG-Grasp dataset.

### A.3.2 EFFICIENCY ANALYSIS

As shown in Fig. 8, we present the AP-time curves of various methods on the TransCG-Grasp dataset. Our approach demonstrates the best balance between performance and running time when compared to existing grasping methods, such as GraspNet BaselineFang et al. (2020), TSB Ma & Huang (2023), GSNet Wang et al. (2021), HGGD Chen et al. (2023), and EconomicGrasp Wu et al. (2024).

### A.4 ADDITIONAL DETAILS FOR REAL-WORLD ROBOT GRASPING

In our real-world robotic experiment, we utilize a Franka Emika Panda robot arm equipped with a Robotiq Adaptive 2F-85 two-finger gripper and an Intel RealSense D435i RGB-D camera, mounted on the end effector. To accurately transform the detected 6-DoF grasping pose from the camera frame to the end-effector frame, we employ the EasyHandeye tool[1], which computes an extrinsic transformation matrix for eye-in-hand calibration.

The robotic system follows a structured pick-and-place pipeline, starting from a predefined initial pose. At this stage, the inference system captures an RGB-D input from the RealSense camera and predicts the optimal 6-DoF grasping pose with the highest grasp score. Motion planning, implemented using Franky[2], is executed in two steps to enhance both precision and safety: first, the end effector moves translationally to align with the predicted grasping point, and then it adjusts its orientation to match the computed 6-DoF grasping pose. Once aligned, the gripper executes the grasp, and the robotic arm returns to its initial pose, ensuring a neutral configuration with zero pitch, yaw, and roll. The object is then placed into a designated container, after which the system resets to its initial state, allowing for continuous execution of the pick-and-place task. The detailed computation process is as follows:

---

[1] https://github.com/IFL-CAMP/easy_handeye
[2] https://github.com/TimSchneider42/franky

$$\begin{aligned}
\mathbf{T}_{\text{grasp}}^{\text{ee}} &= \mathbf{T}_{\text{cam}}^{\text{ee}} \cdot \mathbf{T}_{\text{grasp}}^{\text{cam}} \\
&= \begin{bmatrix} R_{\text{cam}}^{\text{ee}} & t_{\text{cam}}^{\text{ee}} \\ 0 & 1 \end{bmatrix} \cdot \begin{bmatrix} R_{\text{grasp}}^{\text{cam}} & t_{\text{grasp}}^{\text{cam}} \\ 0 & 1 \end{bmatrix} \\
&= \begin{bmatrix} R_{\text{grasp}}^{\text{ee}} & t_{\text{grasp}}^{\text{ee}} \\ 0 & 1 \end{bmatrix} \\
R_{\text{grasp}}^{\text{ee}} &= R_{\text{cam}}^{\text{ee}} \cdot R_{\text{grasp}}^{\text{cam}} \\
t_{\text{grasp}}^{\text{ee}} &= R_{\text{cam}}^{\text{ee}} \cdot t_{\text{grasp}}^{\text{cam}} + t_{\text{cam}}^{\text{ee}}
\end{aligned} \tag{6}$$

where $\mathbf{T}_{\text{grasp}}^{\text{ee}}$ is the transformation matrix of the pose with respect to the end-effector. $\mathbf{T}_{\text{cam}}^{\text{ee}}$ is the extrinsic matrix computed by hand-eye calibration. $\mathbf{T}_{\text{grasp}}^{\text{cam}}$ is the 6D pose inferred from our model. $R$ and $t$ is rotation and translation from the transformation matrix $\mathbf{T}$.

## A.5 LIMITATION AND DISCUSSION

A major challenge in grasping transparent objects is the limitation of traditional depth sensors. Standard RGB-D cameras often struggle to capture accurate data for transparent surfaces, resulting in noisy or incomplete depth information. To overcome this challenge, it is crucial to combine multiple sensor types, such as RGB-D cameras, infrared sensors, and stereo cameras, to improve depth estimation for transparent objects. Additionally, simulated data plays a key role in modeling the physical properties of transparent materials, such as refractive indices, to enhance the model's ability to understand how these objects behave under various conditions Tang et al. (2024). Tools like PyBullet, Isaac Gym, and Blender can be used to generate more simulation data providing large-scale synthetic data without the need for physical setups. Moreover, this work employs a simple early fusion approach to leverage multimodal data to improve transparent grasp performance; exploring more effective fusion methods remains a promising direction for future work.

