# OpenReview forum: "Geometry-Aware 6-DoF Grasp Detection in Complex Scenes"
_ICLR.cc/2026/Conference — ICLR 2026 Conference Withdrawn Submission_

### Official Review · Reviewer_GPk1 · 2025-10-26

**Soundness:** 2
**Presentation:** 2
**Contribution:** 2
**Rating:** 4
**Confidence:** 5

**Summary:**

This paper tackles the challenge of transparent object grasping with two key contributions: the TransCG-Grasp dataset, an expanded version of the TransCG dataset that includes grasp annotations, and the GA-Grasp method, a geometry-aware approach for 6-DoF grasp detection. The GA-Grasp framework leverages a modality-aware sparse tensor module, a geometry-aware sparse U-Net, and a grasp head to effectively fuse multimodal input from RGB images, depth maps, and surface normal and predict 6-DoF grasp pose. The main contributions of the paper are as follow:

1.	The TransCG-Grasp dataset for grasp pose prediction of transparent objects.

2.	A GA-Grasp framework that predicts grasp poses based on depth maps, RGB images, and surface normals.

**Strengths:**

1.	The GA-Grasp method is designed to improve grasping for both transparent and general objects and experimental results on the TransCG-Grasp and GraspNet-1Billion datasets achieved the best performance.
2.	The method proposed in the paper has been validated on real-world experiments with promising results. It would be better if the authors could provide some demonstration videos to showcase the results.
3.	This paper constructs the TransCG-Grasp dataset designed for grasp pose detection of transparent objects.

**Weaknesses:**

1.	The method of this paper lacks significant innovation, as its model architecture is nearly identical to that of GSNet. The only notable modifications include a straightforward concatenation of RGB images and surface normals as additional inputs, along with the removal of graspness for identifying graspable points.
2.	The experiments on transCG-Grasp dataset is insufficient, as the majority of the compared methods are tailored for general object grasping. To enhance the credibility of the assessment, it would be beneficial to include comparisons with methods designed for transparent objects, such as ASGrasp[1], GraspNeRF[2], or SwinDRNet[3]
3.	The writing of the paper lacks clarity. For instance, the second paragraph of the introduction, which discusses 6-DoF grasp pose detection, might be more appropriately placed in the related work section. The introduction should instead focus on highlighting the limitations of previous methods and emphasizing the advantages and innovations of the proposed approach. Furthermore, the method section lacks precise description of the modifications made. For example, on line 201, it is unclear how the "refined depth map" is obtained. Similarly, in the "Graspable Point Prediction" section, the supervised labels of graspable points are not explicitly defined.

[1] Shi, Jun, et al. "Asgrasp: Generalizable transparent object reconstruction and grasping from rgb-d active stereo camera." CoRR (2024).

[2] Dai, Qiyu, et al. "Graspnerf: Multiview-based 6-dof grasp detection for transparent and specular objects using generalizable nerf." arXiv preprint arXiv:2210.06575 (2022).

[3] Dai, Qiyu, et al. "Domain randomization-enhanced depth simulation and restoration for perceiving and grasping specular and transparent objects." European Conference on Computer Vision. Cham: Springer Nature Switzerland, 2022.

**Questions:**

1.	In the paper, which specific method contributes the most to the improvement in grasping performance? Based on my previous experiments, performance declines significantly without using graspness. How does the approach proposed in this paper, as demonstrated in Table 2, achieve an increase in grasping performance instead?
2.	Compared to previous depth completion-based approaches, what are the advantages and disadvantages of the method proposed in the article?
3.	The position of grasp poses predicted by the method in the paper are based on the coordinates of the sampled point cloud. If the point cloud exhibits significant positional deviation, it becomes challenging to predict accurate grasp poses. How does the approach address this issue without performing depth completion to mitigate the point cloud offset?
4.	According to the real-world experiment detailed in Appendix A.4, the input is obtained solely from a RealSense camera, which provides RGB-D data. Since surface normals are not a direct output of this sensor, could you please clarify how they were derived or computed in the experimental setup? Additionally, could you elaborate on how the errors in the computed surface normals might affect the performance or success rate in the real-world grasping experiments?

---

### Official Review · Reviewer_XYrS · 2025-10-27

**Soundness:** 2
**Presentation:** 2
**Contribution:** 2
**Rating:** 4
**Confidence:** 4

**Summary:**

The paper studies the task of 6-DoF grasp detection. More specifically, the paper focuses on the transparent objects. To this end, the authors introduce a new dataset to study transparent object grasping. Besides, the authors use an architecture that takes multi-modal input. Experiments on both transparent object dataset and general object dataset show the proposed model has outperformed other baselines. The authors have further conducted real-world experiments, and the model has achieved good performance.

**Strengths:**

1. Good dataset contribution. Transparent objects have been an important problem to address in the grasping. To this end, the authors have contributed a new dataset focusing on transparent object.

2. Model that can grasp transparent object. The authors have therefore developed a model that can grasp transparent object, which is a contribution to the community.

**Weaknesses:**

1. Novelty of method. The ablation of the paper focuses on the ablation of different combinations of different combinations. This does not highlight the novelty of the proposed model, as taking multi-modal input is very common for grasping system.

2. Novelty of data generation pipeline. As the authors have mentioned in the paper, the data generation pipeline is following the previous work. It would be better if the authors can highlight their novelty there.

3. Presentation of visualisations in Figure 3 and Figure 4. In Figure 3 and Figure 4, the authors have shown some visualisations. However, it is difficult to see the objects clearly as they have been obsecured by lots of grasp marks.

4. Real world experiments lack comparison. The authors have conducted real-world experiments to demonstrate the effectiveness of their method in real-world scenarios. However, there is no comparison with previous models, so it is not sure whether the proposed model brings an improvement over the previous baselines in real-world scenarios.

5. Ablation table lacks the combination without depth. In the ablation table, the authors display the combinations without RGB and without normal. If the setting without depth can be added, the ablation table is more thorough and complete.

**Questions:**

I would suggest the authors to address my concerns in the weakness section during the rebuttal.

---

### Official Review · Reviewer_ShUp · 2025-10-29

**Soundness:** 2
**Presentation:** 2
**Contribution:** 2
**Rating:** 4
**Confidence:** 4

**Summary:**

This paper proposes a 6-DoF grasp detection method, GA-Grasp, for transparent object grasp detection. It comprises two key components: a modality-aware sparse tensor module and a geometry-aware sparse U-Net. First, RGB, depth, and surface normal data are input into the sparse tensor module, and then fed into the sparse U-Net for further processing. To train and test GA-Grasp, the authors annotated the TransCG dataset to construct the TransCG-Grasp dataset. Additionally, the GraspNet-1Billion dataset was used for evaluation. Good experiment results are reported.

**Strengths:**

1. To investigate the problem of transparent object grasping, this paper annotates the TransCG dataset, which is helpful for related research.

2. A new grasping method for transparent objects is proposed, consisting of a modality-aware sparse tensor module, a geometry-aware sparse U-Net, and a grasp head.

3. The paper conducts a set of experiments on the TransCG-Grasp and GraspNet-1Billion datasets to evaluate the proposed method, and reports good performance.

**Weaknesses:**

1. The technical contributions of the modality-aware sparse tensor module and the geometry-aware sparse U-Net are unclear. It is also unclear why combining these two modules, the method can handle transparent objects.

2. The paper needs more in-depth analysis of related works on transparent object grasping, as well as how the proposed method advances this field. The current version is a bit unclear in this regard.

3. The presentation of this paper should be improved. For instance:

* GA-Grasp denotes a method, while TransCG-Grasp denotes a dataset, which is confusing.

* Some sentences are confusing, such as those in lines 80-82.

* In line 254, BCEWithLogitsLoss and CrossEntropyLoss seem to be function signatures in the PyTorch library, but they are not standard English terms.

**Questions:**

1. It is better to add more design rationales and analyze the method in depth to emphasize the technical contributions.

2. It is better to analyze the unique challenges of transparent object grasping in depth, as well as why the proposed method can solve them.

3. It is unclear why the geometry-aware is unique and important, as I think most existing methods are also geometry-aware.

4. Limitations of this work should be discussed.

5. What is IR in line 78?

---

### Official Review · Reviewer_QHxc · 2025-10-31

**Soundness:** 3
**Presentation:** 2
**Contribution:** 2
**Rating:** 4
**Confidence:** 4

**Summary:**

This paper introduces TransCG-Grasp, a grasp-annotated extension of the real-world transparent object dataset TransCG, and proposes GA-Grasp, a geometry-aware 6-DoF grasp detection framework. GA-Grasp integrates three main components: a modality-aware sparse tensor module which fuses RGB, depth and surface normals into sparse tensors, a geometry-aware sparse U-Net (sparse encoder-decoder with pruning layers to keep graspable points), and a grasp head that regresses final 6-DoF grasps from sampled seed points. GA-Grasp achieves a +10.06% AP improvement over the prior best method on TransCG-Grasp, and demonstrates real-robot success rates of 82.0% for transparent objects and 90.6% for general objects (both 100% scene completion). Extensive comparisons and ablations on GraspNet-1Billion show that proposed method GA-Grasp yields good results. The authors also present efficiency analyses and discuss limitations.

**Strengths:**

-- This paper improves a real-world grasp-annotated TransCG-Grasp dataset based on TransCG for transparent objects that fills an important gap.

-- The paper provides a complete and well-recorded real-world robot experiment video, illustrating the method's practicality and usability in real-world scenarios.

**Weaknesses:**

-- As the paper states, "An ideal framework for transparent object grasp detection should be optimized end-to-end for better performance and efficiency", however, the benchmarkvevaluation does not include comparisons with more advanced depth-estimation/reconstruction-
based methods beyond TransCG. For the TransCG specifically, we know that the grasp detector is GraspNet-1Billion and the related architecture in this work is derived from GSNet, so a direct comparison does not by itself establish the superiority of the proposed end-to-end approach.

-- The ablation study indicates that adding surface normals yields only a marginal improvement in grasping accuracy for transparent objects (from 29.24 to 29.73), and there is no ablation examining the three-level supervision for graspable point prediction.

-- The real-robot experiments only evaluate the proposed method and do not include comparisons against other baseline methods.

**Questions:**

-- GraspNeRF [1] integrates reconstruction and grasping into an end-to-end framework for transparent objects — this approach needs more in-depth discussion and direct comparison.

-- The Average column in Table 3 is missing the "AP" label and should be made consistent with the other columns.

-- For the real-robot experiment protocol, I suggest first moving to a pre-grasp pose that has the same rotation as the target grasp pose — this would be more scientifically rigorous.

-- How is grasp-point supervision realized? Is the graspness score thresholded into a binary label, or is a different supervision scheme used? This section requires a precise and clear description.

[1] Graspnerf: Multiview-based 6-dof grasp detection for transparent and specular objects using generalizable nerf. In ICRA, 2023.

---

### Note · Authors · 2025-11-13

I have read and agree with the venue's withdrawal policy on behalf of myself and my co-authors.